



# Development of A Land-River-Ocean Coupled Model for Compound Floods Jointly Caused by Heavy Rainfalls and Storm Surges in Large River Delta Regions

**Anyifang Zhang[1] and Xiping Yu[2]**

[1] Department of Hydraulic Engineering, Tsinghua University, Beijing, China.

[2] Department of Ocean Science and Engineering, Southern University of Science and Technology, Shenzhen, China.

**Correspondence**: Xiping Yu (yuxp@sustech.edu.cn)

**Abstract.** Simultaneous or sequential occurrence of different flood processes, including extreme

storm surges and heavy precipitation, tends to trigger compound floods, which are often destructive to life and property. However, numerical models that fully represent the effect of various flood processes and their interactions have not yet been firmly established. In this study, a coupled land-river-ocean model is developed that considers storm surge, storm wave, astronomical tide, river flow, and precipitation. The coupled model is applied to the simulation of compound

floods induced by tropical cyclones in the Pearl River Delta. The numerical results are shown to agree well with observations on river flow, ocean surface elevation, and inundation area. An attribution analysis implies that contributions from land, river, and ocean processes are usually all important in a compound flooding event. The completeness of the coupling method significantly affects the numerical accuracy.



## 1 Introduction

In major river deltas, devastating floods frequently result from simultaneous or sequential occurrence of multiple events, including severe storm surges and heavy regional rainfalls (Zscheischler et al., 2020). The synergistic impact of multiple events may substantially amplify
the spatial extent and time duration of inundation, resulting in more severe damages than a linear addition of the damages caused by each contributing factor. The problem is further exacerbated since climate change leads to an increase in the temporal and spatial frequency of extreme flooding events (Wahl et al., 2015), while rapid urbanization of major river deltas (Chan et al., 2021) results in the annual escalation of losses due to compound floods.

The study of compound floods has attracted the increasing attention of the scientific community in recent decades. Considerable research efforts have been devoted to elucidating the statistical dependencies among the various mechanisms of flood events, which are crucial to risk assessment. Wahl et al. (2015) reported the temporal variability in the dependence between storm surge and precipitation for the coastal cities in America. Their findings indicated a significant
increase in compound flooding events over the past century, as evidenced by rising Kendall correlation coefficients. Moftakhari et al. (2017) evaluated the bivariate return period of the sea level and river discharge in the future scenario, which indicated that both the failure probability and degree of flood drivers will likely worsen due to global warming.

A close relationship between compound floods and tropical cyclone events has been widely
recognized (Wahl et al., 2015; Fang et al., 2021; Lai et al., 2021; Hendry et al., 2019). Tropical cyclones (TCs), often characterized by a simultaneous occurrence of heavy rainfalls, storm surges, and storm waves, are the most typical weather systems causing compound floods. In a TC-contributed compound flood, none of the component events may have reached their extreme conditions, but their interdependent occurrence can be historically disastrous.

Simulation of a TC contributed compound floods in a major river delta region requires coupled models. An ocean circulation model can be used to describe the ocean surface elevation and the ocean flow jointly caused by storm surges and astronomical tides; an ocean wave model





is able to predict the wind wave spectrum; a river flow model usually results in the water level and the flow rate within the river channel; a hydrologic model may be used to represent the rainfall-runoff process. When all or part of these models are integrated into a system, the framework of a land-river-ocean coupled model becomes available. The degree of coupling determines whether the synergistic effect of multiple events can be reasonably obtained. A properly coupled model system can simulate different kinds of compound floods as long as the atmospheric forcing can be provided (Gori et al., 2020b; Feng et al., 2022; Revel et al., 2023; Xu et al., 2023; Du et al., 2024; Zhong et al., 2024). Lee et al. (2019) proposed a coupled model for TC landfall in Korea, highlighting the importance of the rainfall-runoff process in studying the inundation. Gori et al. (2020a) coupled the hydrological model with river and ocean dynamic models to investigate the compound flood induced by six TCs in the Cape Fear Estuary, with an emphasis on the effect of rainfall structure on compound floods. Most of the existing models, however, are oversimplified in some parts or limited in the coupling degree. Particularly, an accurate estimation of the air-sea momentum exchange under extreme wind speed (Zhang and Yu, 2024) has not been taken into consideration when modeling TC induced compound floods.

In this study, a land-river-ocean coupled model is developed, which can comprehensively describe the dynamic details of storm surge, storm wave, astronomic tide, river flow, inundation, and precipitation, as well as their interactions (**Figure 1**). The atmospheric wave boundary layer model is employed to improve the accuracy of the atmospheric forcing on the ocean. The coupled model is then applied to the simulation of TC induced compound flood in the Pearl River Delta. The computed water surface elevation, river discharge, and inundation areas during typical TC events are satisfactorily verified by measured data.



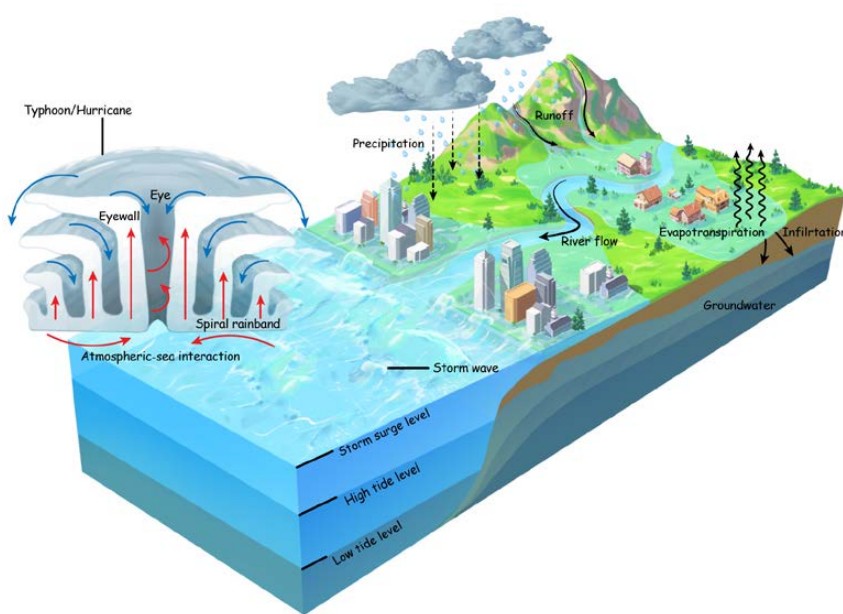


**Figure 1.** Physical processes represented in the land-river-ocean coupled model.

## 2 Model Integration

### 2.1 General Description

The coupled model system to be established in this study aims to correctly represent the

temporal and spatial variation of the water surface elevation and flow rate in rivers and the coastal

ocean, and also in the inundation area if overflows occur, which are jointly caused by storm surge,

storm wave, astronomical tide, river flow, and regional rainfalls. The details of the model system,

as well as the method of coupling, are shown in **Figure 2**.

A complete model system may also require a general atmospheric circulation model, usually

called the General Circulation Model (GCM) by meteorologists (Satoh, 2013), so that the wind

flow velocity, atmospheric pressure, precipitation, etc., can be numerically determined. In the

present study, however, the atmospheric forcing is directly derived from reliable reanalysis data





for a past event and may have to be obtained with numerical weather prediction for a coming event. For a future scenario, we may rely on an appropriate long-term climate model.

As a basic feature, we require that the model we established can resolve the instantaneous water surface elevation and flow rate caused by both astronomical tides and storm surges. The astronomical tides are considered to be oscillations forced at open boundaries. The storm surges are jointly caused by the wind shear at the air-water interface, the air pressure acting on the ocean surface and the effect of the radiation stresses originating from the ocean surface waves. The storm

waves should also be paid attention to not only because they contribute to the mean water level variation but also because they directly cause a significant elevation of the water surface. The storm-wave induced water flows are not fully resolved in our model because their contribution to the wave-filtered water level is of a second order so that it is enough only to include the effect of wave radiation stress (Longuet-Higgins and Stewart, 1964; Dietrich et al., 2012). Since the wind

stress on the free water surface is an important parameter for both storm surge and ocean wave modeling, an enhanced atmospheric wave boundary layer model is necessary considering the condition of strong wind and shallow water (Zhang and Yu, 2024).

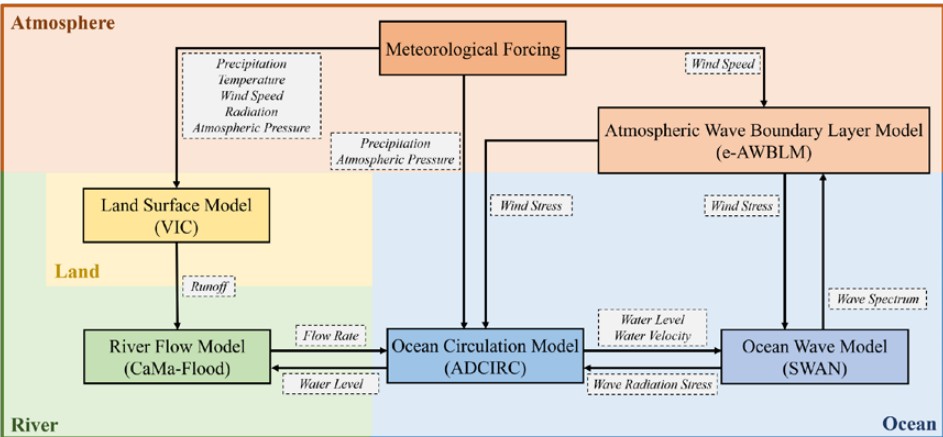

**Figure 2.** Flowchart of the coupled model for compound floods.



We also require that the model we established can accurately describe the physical process of the flow routing in the river system, consisting of main streams and their tributaries, which receive runoff generated by a land surface model. The flow routing model, with which the water depth and the flow discharge must be determined, may be based on the governing equations for unsteady open channel flows, or a significantly simplified conceptual model preferred by applied hydrologists. The land surface model must be able to yield the runoff given the precipitation and other necessary parameters related to atmospheric forcing, land cover, and soil properties.

## 2.2 Components of Model System

**Ocean Circulation Model.** Advanced CIRCulation (ADCIRC; Luettich et al. (1992)), solving the two-dimensional shallow water equations, is adopted to determine the water surface elevation and the vertically-averaged flow rate in the ocean. The shallow water theory assumes that the water depth is much smaller than the length scale in the horizontal directions of the problem. Therefore, ADCIRC can represent only long waves such as astronomical tides and storm surges, but not wind waves. The contribution of precipitation has been considered as a source term in the mass conservation equation (Bilskie et al., 2021). The astronomical tides induced water surface elevation is forced at the otherwise undisturbed open boundaries. At the boundary where the ocean circulation model and the rive flow model match, smooth water surface, and continuous flow rate are required. Along the coastline, free run-up conditions are specified at beaches while no-penetration conditions are given at seawalls.

**Ocean Wave Model.** Simulating WAves Nearshore (SWAN; Booij et al. (1999)) is utilized to predict the evolution of the phase-averaged wave energy spectrum, from which the wave height can be evaluated in a statistical sense. The governing equation of the model is based on the conservation of wave action, which is generalized from the conservation of wave energy when there is a steady current at present. Wind energy input, wave energy dissipation, and wave energy redistribution due to nonlinear wave-wave interactions are treated as sources for wind wave development.



**Atmospheric Wave Boundary Layer Model.** The enhanced Atmospheric Wave Boundary Layer Model (eAWBLM; Zhang and Yu (2024)) is employed to estimate the wind stress acting on

the ocean surface, which is an indicator of the intensity of momentum transfer through the air-sea interface and an important parameter in both the ocean circulation model and the ocean wave model. The model is essentially based on the momentum and energy conservation within the atmospheric wave boundary layer over the ocean surface. It was recently improved to correctly describe the effect of wave breaking under very strong wind conditions and also the effect of finite

water depth (Chen and Yu, 2016; Xu and Yu, 2021; Zhang and Yu, 2024).

**River Flow Model.** Catchment-based Macro-scale Floodplain (CaMa-Flood; (Yamazaki et al., 2011)) is chosen to determine the flow rate and water depth in the river system, consisting of a mainstream and its tributaries. The model is based on a significantly simplified form of the basic equations for open channel flows in order to achieve a high computational efficiency. The lateral

inflow is given by a land surface model. The river mouth is connected to the ocean and the matching boundary conditions must be satisfied. If truncated at any place, an inflow condition, called river base flow, must be prescribed at the upper end of the mainstream. A very important advantage of CaMa-Flood, as compared to many other river flow models, is that inundation can be simulated.

**Land Surface Model**. Variable Infiltration Capacity (VIC; Hamman et al. (2018)), a distributed macroscale hydrologic model, is employed to estimate the runoff into the river system. The model takes into consideration key hydrological processes, including evaporation, infiltration, moisture movement, and runoff generation. With known meteorological forcings, the surface runoff and the baseflow are evaluated based on the variable soil moisture capacity curve (Liang et

al., 1994) and the Arno model (Franchini and Pacciani, 1991), respectively.

**3 Model Application in the Pearl River Delta Region**

In this study, we perform hindcasting of five tropical cyclone (TC) events [Hagupit (2008); Koppu (2009); Vicente (2012); Hato (2017); Mangkhut (2018)], which caused destructive floods



in the Pearl River Delta during the past two decades, to test the validity of the model we developed

to simulate compound floods. The landfall intensity of Koppu (2009) is classified as a Typhoon

(TY) and the other tropical cyclone events are classified as Severe Typhoon (STY) according to

the China Meteorological Administration (Lu et al., 2016). It is worthwhile to mention that all five

TC events made landfall at the southwest part of the Pearl River Delta, with the right-front quadrant,

i.e., the prolonged stronger wind and lower pressure condition, covering the area under our

consideration (**Figure 3a**).

The boundary between rivers and the ocean is set at the cross-sections of the major tributaries

of the river system where the 10 m topographic contour crosses. In the Pearl River Delta region,

there are 7 such cross-sections, from which the discharge accounts for more than 97% of the

rainfall-runoff generated in the entire catchment  (**Figure 3b**).

**3.1 Discretization of the River System**

We discretize the 7 sub-catchments, each with over 2,000 square kilometers of drainage area

controlled by the cross-section where a river flows into the ocean (**Figure 3b**). In the VIC model,

the soil with a 3-layer structure and the surface vegetation are parameterized with the

OpenLandMap (Tomislav, 2018) and the Global Land Cover Facility (GLCF; Hansen et al. (2000))

respectively. The soil properties, including infiltration capacity, saturated hydraulic conductivity,

bulk density, and wilting point, are estimated according to the soil type (Twarakavi et al., 2010;

Cosby et al., 1984). A database on global river width for large rivers is used to estimate the width

of large rivers, i.e., river width larger than 300 m in this study (Yamazaki et al., 2014). For small

rivers, the river width and the river depth are estimated with empirical formulas (Yamazaki et al.,

2011). The meteorological forcings, including precipitation, wind speed, temperature, surface

radiation, pressure, and humidity, are obtained from the ERA5-land dataset (Hersbach et al., 2020).

Square grid cells are adopted for both VIC and CaMa-flood. The total number of land surface cells

for VIC is 6141 with a spatial resolution of 5′ while the total number of river channel elements for



CaMa-flood is 14397 with a spatial resolution of 3′. The computational time step is set to 1 hour

in VIC and 10 minutes in CaMa-flood, respectively.

## 3.2 Discretization of the Coastal Ocean

An unstructured mesh covering the Pearl River Delta region, which consists of very complex river networks, is carefully built (Roberts et al., 2019; Qiu et al., 2022). The land-ocean boundary extends from the coastline to the inland location where the 10 m contour reaches in order to fully

include the floodplain (**Figure 3c**). The Forest And Buildings removed Digital Elevation Model (FABDEM; Hawker et al. (2022)), which removes trees and buildings to represent bare-land terrain, is utilized to determine the land elevation. The mesh resolution along the river and its tributaries is refined to 50 m to resolve storm surge propagation within the river system and inundation over the land (**Figure 3d**). The computational domain consists of 1,413,038 elements

and 721,704 nodes. The bottom friction of the land region is estimated based on Manning friction law. In the ocean circulation model, spatial variations of the hydraulic roughness are considered based on the variability of the land cover type (Mattocks and Forbes, 2008; Yang and Huang, 2021). The spatial variation of the land cover type and the corresponding value of the Manning coefficient are shown in **Figure S1** and **Table S1**. Note that the 7 inflow boundaries in the ocean

circulation model are also the outflow boundaries of the relevant sub-catchment in the river flow model. The computational time step for coupling at the river-ocean confluences is set to 1 hour. The astronomical tide is forced hourly by the water surface elevation at the open boundary (Egbert and Erofeeva, 2002). Moving boundaries in ADCIRC are treated with the conventional dry-and-wet approach. For the numerical stability, the drying and wetting threshold is set to 0.1 m and the

computational time step is set to 1 second in ADCIRC. SWAN is dynamically coupled with ADCIRC every 10 minutes. In SWAN, the frequency range is set to 0.0157–1.57 Hz and the directional resolution is set to 10°.



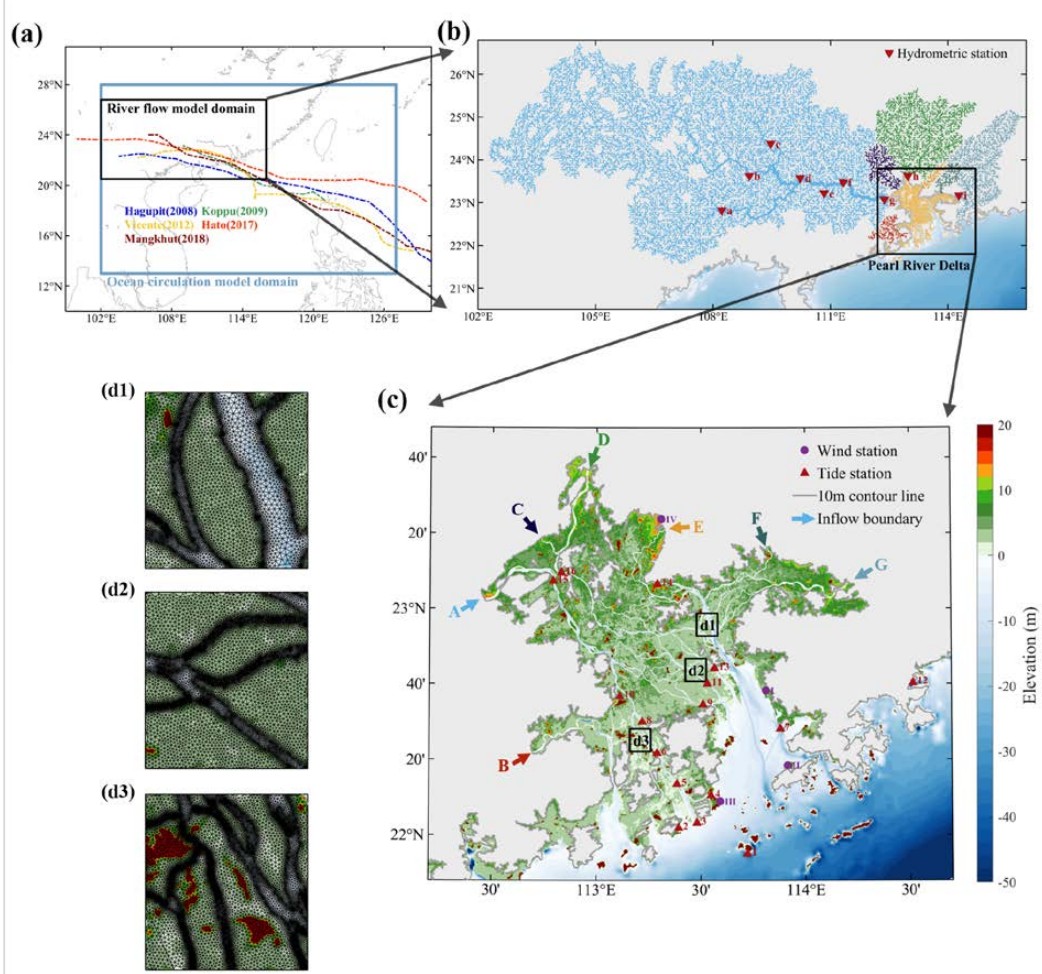

**Figure 3.** Detailed description of the compound flood model applied in the Pearl River Delta. **(a)** The computational domain and the tracks of selected tropical cyclones. **(b)** The river network in the seven sub-catchments of the Pearl River basin. The inverted red triangles marked with a-i are the locations of discharge stations. **(c)** Elevation of Pearl River delta region used in the ocean circulation model. The red triangles marked with 1-16 are the locations of tide stations. The purple circles marked with I-IV are the locations of meteorological stations. The arrows marked with A-G are the inflow boundaries. The color of the river networks in (b) corresponds to the inflow boundaries in (c). **(d)** Enlarged view of refined mesh along the river channel.



Parametric models are employed to determine the wind velocity and the air pressure within a circular area surrounding the TC center obtained from the best track dataset (Lu et al., 2016). More specifically, the wind velocity and the air pressure are computed using the empirical models proposed by Emanuel and Rotunno (2011) and Holland (1980). The radius of max wind speed in the models is estimated with the formula proposed by Willoughby and Rahn (2004). When applied to the ocean circulation model and ocean wave model, the inflow angle (Bretschneider, 1972), the translational velocity of moving TCs (Jelesnianski, 1966), the spatial conversion factor (Georgiou et al., 1983), and the time conversion factor (Powell et al., 1996) are involved to obtain the 10 m-10 min (10 m above the mean sea level and 10 min average) wind velocity field. Far away from the TC center, the meteorological forcings are derived from the reanalysis data in ERA5 (Hersbach et al., 2020). The eventual forcing fields are then weighted by the empirical results and reanalysis data in terms of the distance from the position of interest to the TC center (Carr Iii and Elsberry, 1997). Detailed information on the construction of the meteorological forcings due to the presence of tropical cyclones can be referred to previous studies (Lin et al., 2012; Yang et al., 2019; Xu and Yu, 2023; Zhang and Yu, 2024). The wind velocity fields are validated as demonstrated in **Figures S2-S6**.

## 4 Numerical Results

### 4.1 River discharge

The river flow model is validated by comparing the computed and measured daily discharge at the nine hydrometric stations (**Figure 4**). The computed results show satisfactory agreement with measured data in general, with a Wilmott Skill level of 0.960. Exceptions are noted at the upstream stations such as Qianjiang and Liuzhou during the dry season. The discrepancy is considered to be caused by an omission of the reservoir operation at further upstream of the river. It is worthwhile to emphasize that the model is capable of capturing the peak discharge events rather accurately.



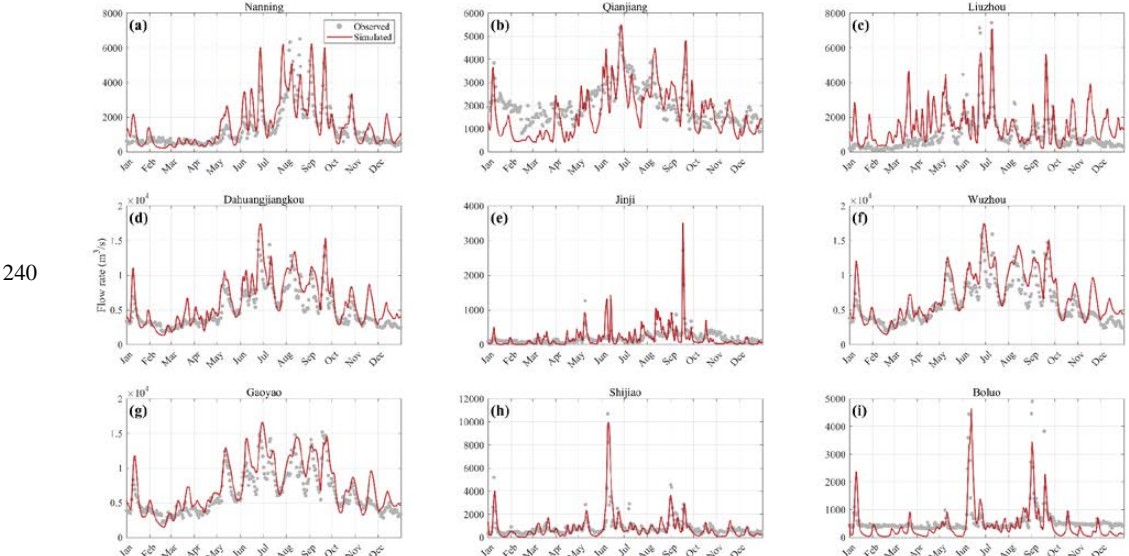

**Figure 4.** The simulated (red lines) and observed (gray points) river flow rate in 2018 are compared at different stations.

## 4.2 Storm Surge

The computed water level elevation due to storm surges is compared with the observations at the tide gauge stations, as shown in **Figure 5**. The agreement between computational and observational results is very good in general. Some discrepancy at particular places is known to be an effect of mismatch between available topographic data and the real situation due to human activities (Zhang et al., 2021). It is worthwhile to mention that all storm surges nearly coincided with the high tide level, especially during Typhoon Mangkhut (2018) and Typhoon Hagupit (2008), since the maximum surge levels are the major concern from the disaster prevention point of view. The agreement between the computed and observed maximum surge levels is particularly good, with a high Wilmott Skill level of 0.887. We may also have to mention that our coupled model overestimates the maximum surge level by about 0.5 m at stations 15 and 16 during Typhoon Hato (2017). Even so, our present results are significantly better when compared to the previous ones which overestimated by about 2 m (Qiu et al., 2022; Zhang and Yu, 2024). Previous



overestimations are partially due to an incomplete coupling between river and coastal flows and
partially due to an omission of possible inundation.

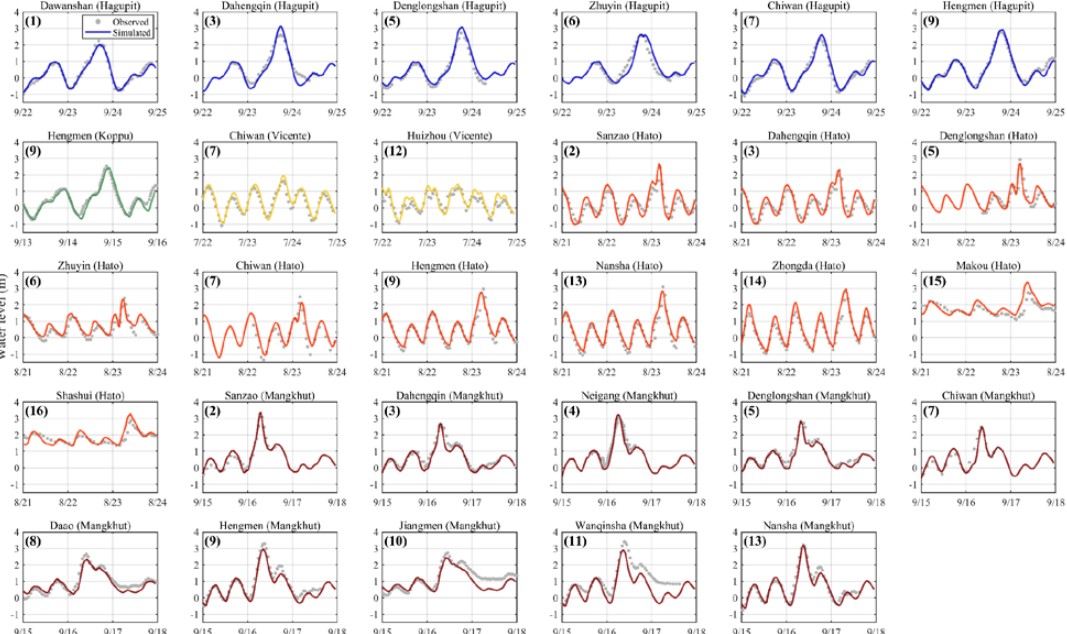

**Figure 5.** The simulated (solid lines) and observed (gray points) storm tides are compared for
selected Typhoons.

## 4.3 Inundation

We employed the daily MODerate resolution Imaging Spectrometer (MODIS,
https://ladsweb.modaps.eosdis.nasa.gov) dataset, with a resolution of 500 m, to identify inundation
areas during each Typhoon event. 7 days, starting from the landfall time of each Typhoon, is
selected for each event to ensure a comprehensive assessment of the inundation extent. Despite of
some problems caused by cloud cover, remote sensing techniques have been widely used for flood
identification and have rather successfully captured the maximum extent reached by the highest



water levels (Brakenridge et al., 2013). The identification approach and threshold utilized in this

study followed Tellman et al. (2021).

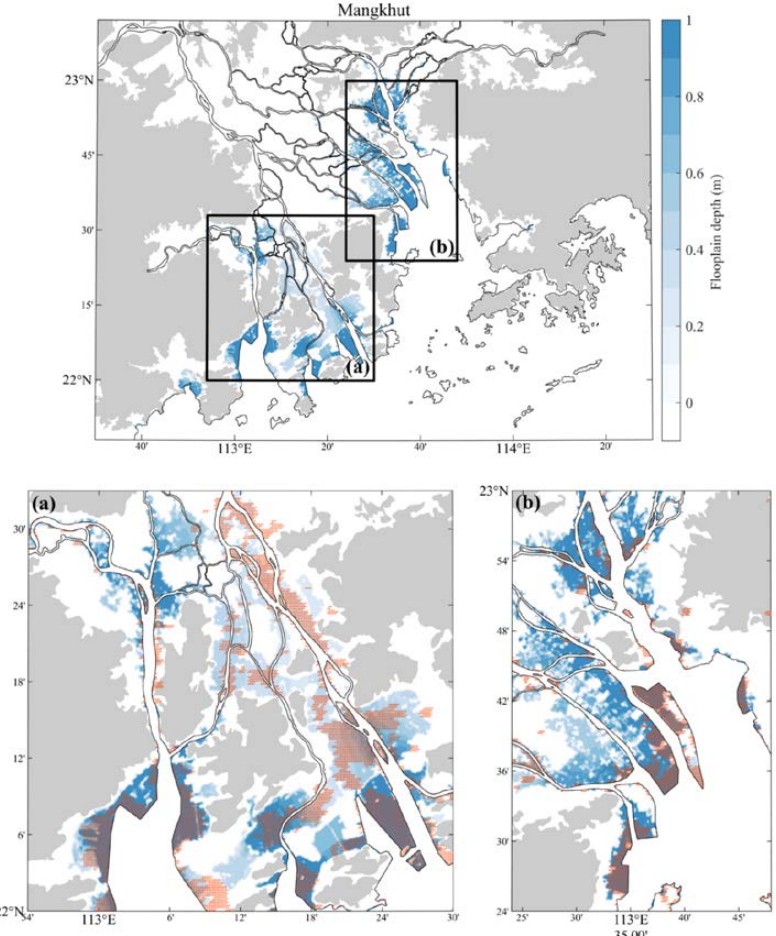

**Figure 6.** The contour plot (blue) of simulated inundation depth in Typhoon Mangkhut. The orange dots show the inundation range estimated by MODIS datasets.

The computed inundation area during Typhoon Mangkhut (2018) is compared with the

satellite results derived from MODIS, as presented in **Figure 6**. Comparisons for other Typhoon

events are demonstrated in **Figures S7-S10**. Since the cloud cover removing techinique in Tellman

et al. (2021) method is known to result in an underestimation of the flooding area, it is reasonable





to expect that the computed inundation range exceeds the observed area to some degree, as shown

in **Figure 6**. Certainly, the numerical model leads to satisfactory results on inundation areas.

**5 Discussions**

**5.1 Attribution analysis of compound floods**

It may be useful to understand the attribution of the land, river, and ocean processes to a compound flood in the Pearl River delta region (**Figure 3c**). For this purpose, two additional

scenarios are computed for each Typhoon case, i.e., ocean processes plus river base flow only and ocean processes plus precipitation only. To isolate the attributions of precipitation and river base flow, we can thus subtract the results of these two scenarios from the inclusive results (ocean processes plus river base flow and precipitation). The residual may then be attributed to the ocean processes, which include storm surges, storm waves, and astronomic tides. It is worth

acknowledging that the nonlinear interactions among the various contributing factors are completely neglected in such an approach (Bilskie and Hagen, 2018). Nonetheless, it is still reasonable if we consider the ocean processes as the dominant contributor to the compound flood and discuss the additional attributions of precipitation and river base flow to the ocean processes.

The spatial distributions of the inundation depth due to the ocean processes, the river base

flow, and the precipitation during Typhoon Mangkhut (2018) are presented in **Figure 7**. The distribution of the inundation depth during other Typhoon events can be found in **Figures S11-S14**. It is demonstrated that the ocean processes, as anticipated, cause inundation near the coastline. In most cases, the inundation areas due to ocean processes account for over 90% of the total (**Figure 7a**). River base flow, on the other hand, plays an important role in the upstream regions

along the river channels (**Figure 7b**). Precipitation affects a broader area but with some concentrations in locally lower inland areas (**Figure 7c**).

These distribution characteristics remain almost the same for all Typhoon events, although some quantitative discrepancies do exist among different events. For instance, Typhoon Hagupit (2008), which was more significantly affected by the ocean process, caused heavy flooding in the





coastal regions (**Figure S11**), while Typhoon Hato (2017), which occurred when the river base

flow was considerably strong, resulted in a large inundation area in the inland regions (**Figure S14**).

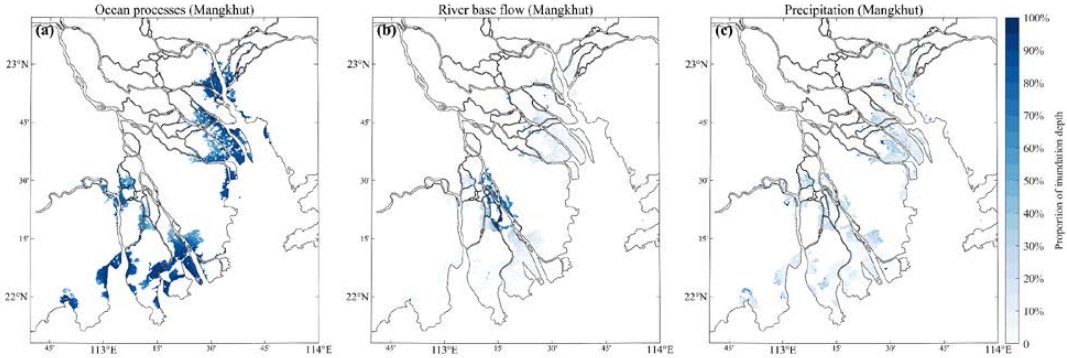

**Figure 7.** The spatial distribution of inundation depth due to (a) ocean processes, (b) river base flow, and (c) precipitation during Typhoon Mangkhut (2018).

To further quantify the attribution of each factor, we estimate the inundation volume due to the precipitation, the river base flow, and the ocean processes, based on numerical results presented in **Figure 7** and **Figures S11-S14**. For all Typhoon events under our consideration, the flooding

volume caused by precipitation is relatively small, ranging from 5% to 15%. In contrast, the ocean processes are responsible for more than half of the flooding volume. Inundation caused by river base flow varies from approximately 30% during Typhoon Vicente (2012) and Typhoon Hato (2017) to around 2% to 10% during Typhoon Hagupit (2008), Typhoon Koppu (2009), and Typhoon Mangkhut (2018) (**Figure 8**). Note that Typhoon Vicente (2012) and Typhoon Hato

(2017) made landfall on 23 July and 22 August, when the daily average discharge of river base flow was 10,975 $m^3$/s and 13,389 $m^3$/s, respectively. While Typhoon Hagupit (2008), Typhoon Koppu (2009), and Typhoon Mangkhut (2018) made later landfall on September 23, 14, and 16, when the daily average river discharge was 2,967 $m^3$/s, 2,829 $m^3$/s, and 7,172 $m^3$/s, respectively.



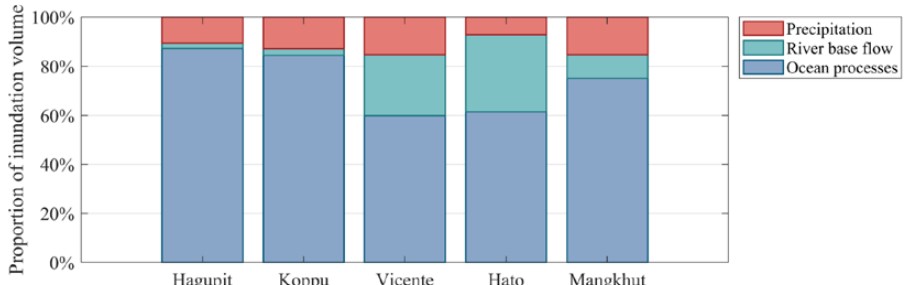

**Figure 8.** The attribution of flooding volume due to ocean processes, river base flow, and precipitation during disastrous Typhoon events.

## 5.2 Comparison between two-way and one-way coupling

Existing coupled models for simulating compound floods often adopt the one-way coupling approach, i.e., the river flow model transfers information to the ocean circulation model without receiving feedback (Deb et al., 2023; Gori et al., 2020b; Du et al., 2024; Bakhtyar et al., 2020). To illustrate the differences between the two-way and one-way coupling between the river flow model and ocean circulation model, the discharge and water level at cross-section A, where the largest sub-catchment meets the ocean (**Figure 3b** and **Figure 3c**), are compared in **Figure 9**. The actual discharge shows rhythmic fluctuations due to the influence of astronomic tides, which are totally omitted in the one-way approach. This may lead to an underestimation of the extent of inland inundation resulted from the river flow model on some occasions. The compound flood induced by the ocean processes and rainfall-runoff resulting from the river flow model is shown in **Figure 10** and **Figures S15-18**, where the inundation area is mainly located along the river channels. Compared to the two-way model, the one-way coupled model significantly underestimates the inundation area, especially near the boundaries where rivers meet the ocean. It may be interesting to note that the ocean surface oscillations affect a longer distance of the tidal river when the river flows are relatively weak (**Figure 10**, **Figures S15**, and **Figures S16**) and vice versa (**Figures S17** and **Figures S18**).





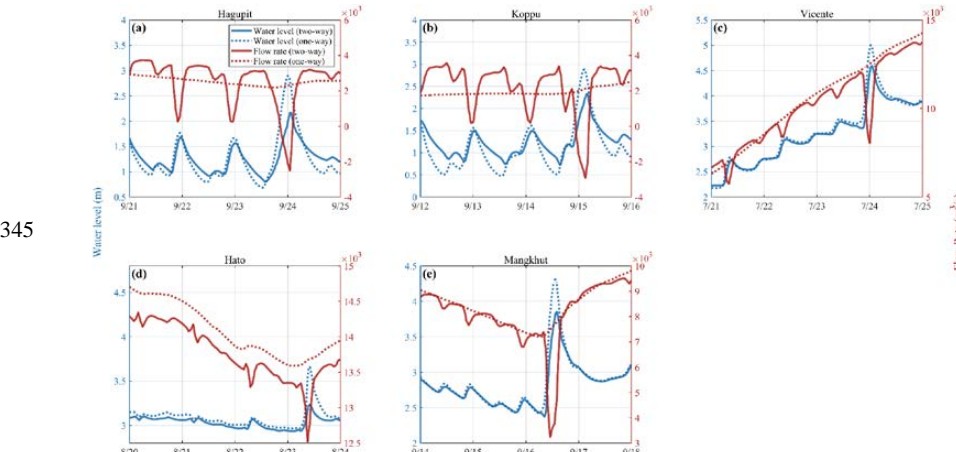

345

**Figure 9.** Water level and flow rate at coupled boundary A (Figure 3c). The solid lines and dashed lines are the results from two-way and one-way coupling approaches, respectively.

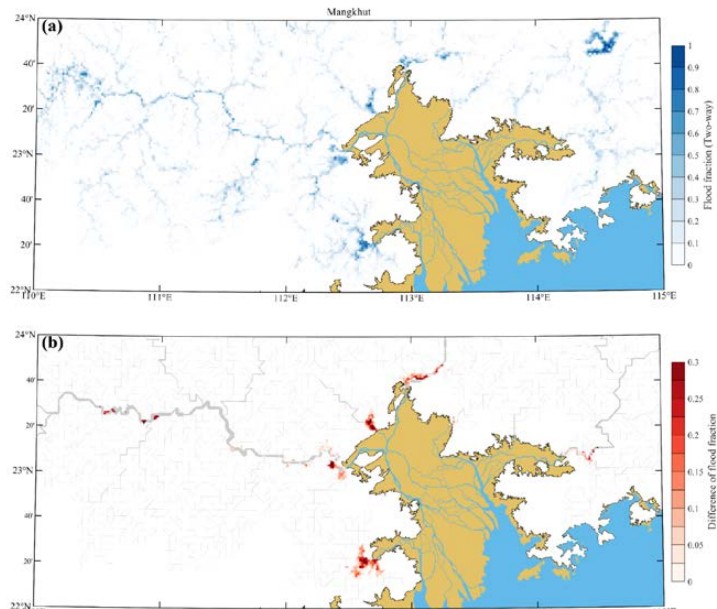

350     **Figure 10. (a)** The fraction of flooding area during Typhoon Mangkhut (2018) estimated by the CaMa-flood model with two-way coupling approach. (b) The difference of the fraction of flooding area between the two-way and one-way coupling approaches. The river channels are marked with gray lines, with the width of the river channel indicated by the line thickness.



## 6 Conclusions

A land-river-ocean coupled model is developed in this study for simulating compound floods, considering the possible effect of storm surges, astronomical tides, storm waves, precipitation, and river flow. The model is applied to the hindcasting of the compound floods induced by five Typhoon events occurred in the Pearl River Delta region and is shown to perform well. Based on numerical results from the model we developed, we are able to discuss the attribution of land, river, and ocean processes in TC contributed to compound floods that occurred in the Pearl River delta region. The ocean processes are demonstrated to be the dominant contributor in all events, while the land and river processes also play important roles. It is shown that the one-way coupling approach results in an underestimation of the inundation area given by the river flow model owing to the absence of feedback from the ocean circulation model.

*Code availability.* All models employed in this study are available as open source. The source code of eAWBLM can be downloaded from https://github.com/anyifang/e-AWBLM. The source code of coupled SWAN and ADCIRC can be requested at https://adcirc.org. The source code of Cama-Flood can be downloaded from http://hydro.iis.u-tokyo.ac.jp/~yamadai/cama-flood. The source code of VIC can be downloaded from https://github.com/UW-Hydro/VIC.

*Data availability.* Meteorological forcings on ocean and land were obtained at https://cds.climate.copernicus.eu/cdsapp#!/dataset/reanalysis-era5-single-levels?tab=overview and https://cds.climate.copernicus.eu/cdsapp#!/dataset/reanalysis-era5-land?tab=overview, respectively. Tropical cyclone parameters were obtained at https://tcdata.typhoon.org.cn/. Bathymetry was obtained at https://download.gebco.net/. Land elevation was obtained at https://data.bris.ac.uk/data/dataset/25wfy0f9ukoge2gs7a5mqpq2j7. The land cover type was obtained at https://zenodo.org/records/5210928. Surface vegetation type was obtained at http://app.earth-observer.org/data/basemaps/images/global/LandCover_512/ LandCoverUMD_512/LandCoverUMD_512.html. Soil type was obtained at



https://zenodo.org/records/2525817. Tide information was obtained at

https://g.hyyb.org/archive/Tide/TPXO/TPXO_WEB/

*Competing interests.* The contact author has declared that none of the authors has any competing
interests.

*Acknowledgments.* We thank Prof. Dai Yamazaki for installing the Cama-flood model. We also
thank Dr. Yue Xu and Prof. J C Dietrich for their support on running coupled SWAN and
ADCIRC.

*Financial support.* This study is financially supported by the National Natural Science
Foundation of China under Grant No. 41961144014.

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
