# Peer review of "Development of A Land-River-Ocean Coupled Model for Compound Floods Jointly"

_EGUsphere, 2024_

## Author Response (AR1)

**Authors' Responses to Comments of Reviewer #1**

*We appreciate very much the constructive comments and suggestions of the reviewer and have revised the manuscript accordingly. In the following, we explain our response to each comment of the reviewer. All revisions are highlighted in red color in the marked manuscript.*

**General Comments:** This work develops a Land-River-Ocean coupled model and evaluates its performance in simulating flood inundation through its application in the Pearl River Estuary. The results indicate that the model's simulation matches well with observations, demonstrating the potential practical application value of the developed model. Overall, this work leans more toward technical model development and evaluation, with relatively weaker scientific novelty. However, considering its practical application value, I recommend publication after revisions. Specific comments are as follows:

*Response:*

Thank the reviewer for his/her general comments on the practical value of our numerical model. Existing coupled models often oversimplify some dynamic processes and their interactions that have significant effects on flooding phenomena in order to save computational costs. In this study, we comprehensively account for air-sea interactions as well as the interactions between river floods and storm surges. We believe that the model presented in this study provides a more physically accurate description of the complex interactions among various flood processes while maintaining a reasonable computation efficiency. In the following, we try to address the reviewer's concerns and improve the manuscript accordingly.

**Comment 1.** Incomplete model configuration information. In Section 3, the authors chose the Pearl River Estuary as the study area and examined several typhoon-related flood inundation events to evaluate the model's simulation capability. However, the authors seem to omit some critical information. For example, the source of depth and elevation data is not shown, which is crucial as they largely determine the simulation results. Did the authors use publicly

available topographical data, or did they extract it from nautical charts? What is the resolution? How accurate is it? These details should be explained thoroughly in the text. The source of the boundary conditions and tidal forcing data for the ocean model. What is the resolution of these data? The authors also seem to have omitted this information entirely.

***Response:***

We are sorry for omitting some basic information related to model application. In fact, the shoreline of the South China Sea is derived from the Global Self-consistent, Hierarchical, High-resolution Geography (GSHHG), and the outline of the river branch is obtained from the Open Street Map. The bathymetry of the open sea is obtained publicly from GEBCO (GEneral Bathymetric Chart of the Oceans), which provides the global ocean elevation with a spatial resolution of 15′. The bathymetric of Pearl River Delta is obtained from the Pearl River Water Resources Commission with a special permission, which is of a higher resolution (80 m). As for the astronomical tide, which consists of 13 tidal constituents (M2, S2, N2, K2, K1, O1, P1, Q1, MF, MM, M4, MS4, and MN4) from TPXO (Egbert and Erofeeva, 2002), is prescribed hourly as the water surface elevation at the open boundary. We added theses information in the revised manuscript [Page 9, Lines 189-195; Page 11, Lines 218-220; Page 22, Lines 412-425].

**Comment 2.** The model's advantage. The authors frequently mention the advantage of this coupled model in the text. For instance, in Lines 59-69, the authors state that previous models have simplified the coupling to some extent, whereas the model developed here has made no such simplifications. Similarly, in Lines 252-257, the authors claim that the accuracy of their model is higher than that of previous studies. They further suggest that one possible reason for the improved accuracy is the more comprehensive coupling in their model. I believe the highlight of this work is indeed the more comprehensive coupling. However, the authors do not provide any direct evidence to demonstrate that the improved simulation results are due to this comprehensive coupling. Simply comparing the results to previous studies is insufficient. The differences in simulation results could arise from many factors, such as

differences in topographical data or atmospheric forcing data. The authors have not excluded these factors before concluding that the improvements are due to the coupling, which is not convincing.

***Response:***

Thanks for the comment. In the model we developed, the description of the air-sea momentum transfer is really improved. In most of the existing storm surge models, the wind stress coefficient, which is a measure of the air-sea momentum flux, is estimated with empirical formulas proposed decades ago, e.g., the linear formula proposed by Garratt (1977) and Large and Pond (1981). However, most of the earlier formulas have been demonstrated to be incorrect under extreme wind conditions. Recent studies showed that the result of wind stress coefficient from the e-AWBLM agrees well with observations from deep to shallow water (Fig. R1; Zhang and Yu, 2024), and we fully coupled the e-AWBLM into our present model.

[Figure]

Figure R1. The wind stress coefficients at different water depths. Dashed lines with error bars are observations; solid lines are parametric equations and simulation results.

We applied the model we developed to the computation of the maximum surge level during several Typhoon events. In order to exclude the influence of non-model factors, tidal levels without adding the storm surge are well modeled to exclude the effects of topography and bottom friction on water levels (Figure 5). The wind speed and direction are well modeled to exclude the impact of atmospheric forcing (Figure S2-S6). Based on such obtained results,

verification of the effectiveness of the present model should be meaningful. We included simulation results using the default wind stress evaluation method in ADCIRC (Garratt, 1977). Overall, the maximum surge levels obtained with the e-AWBLM are more accurate than those obtained with Garratt (1977)'s formula. It is clear that Garratt (1977)'s formula significantly underestimates the maximum water level.

[Figure]

Figure R2. The simulated maximum surge levels are compared with observations in (a) Garratt (1977) and (b) e-AWBLM

To improve the description of the river-ocean interaction, the two-way coupling of the river model and the ocean circulation model is used in the model we developed. For verification, we compare the water level elevation at Makou (15) and Sanshui (16) stations during Hato (2017). It is clear that the one-way coupling overestimates the maximum surge levels and surge recession due to a fake accumulation of storm water at the inflow boundary (Figure R3). Note that we still overestimate the maximum water level at two stations, probably due to the neglect of some small rivers (rivers with channel width <50m are neglected due to the limitation of the resolution of the grid). We believe that the one-way coupling method overestimates inundation in the upstream region covered by the ocean circulation model because the storm surge wave cannot propagate into the river without a two-way coupling.

[Figure]

Figure R3. The simulated (solid lines) and observed (gray points) storm tides are compared at two upstream stations.

We added some detailed explanations in the revised manuscript based on the reviewer's comments [Page 13, Lines 266-269, Lines 275-278; Page 14, Figure 6; Figure S7 in supporting material].

**Comment 3.** Line 16-18: This statement is obvious. The authors conducted quantitative analyses in the text and should provide their quantitative contributions here.

*Response:*

Thanks for the comment. The numerical results agree well with observations on river flow and ocean surface elevation, with the Wilmott Skill value of 0.96 and 0.88, respectively. In addition, the simulated coastal inundation area covers approximately 80% of the area identified through remote sensing. We included the results of the quantitative analyses in the abstract [Page 1, Lines 15-22].

**Comment 4.** Line 128: When calculating wind stress here, the eAWBLM model considers the influence of waves. Does it also account for ocean currents? Figure 2 seems to suggest that ocean currents are not considered in this model.

*Response:*

Thanks for the comment. The e-AWBLM considers the effect of ocean waves on the transfer of horizontal momentum across the ocean surface, i.e., the wind stress acting on the ocean surface. The ocean currents may affect the ocean waves. This part of the ocean current effect is represented when the ocean wave spectrum is formulated. Another effect of the ocean

current may exist because it changes the relative velocity of the wind. Since the velocity of ocean currents is usually much smaller than the wind speed, particularly under extreme weather conditions, it may be reasonable to omit this effect.

**Comment 5.** Line 137-138: remove the extra parentheses.

*Response:*

Thanks for the comment. We corrected the format of the citation [Page 7, Lines 140].

**Comment 6.** Line 230: In this section, the authors analyze the results for runoff, storm surges, and flood inundation, showing that the model performs well. However, as mentioned in Major Comment 2, the authors should conduct a more in-depth study of what aspects the model performs well in and what aspects it does not. For example, they could conduct sensitivity experiments to demonstrate which aspects are due to the coupling and which are due to accurate wind field data.

*Response:*

Thanks for the comment. As we mentioned when we response to comment 2, we highlighted our improvements in describing the air-sea momentum exchange (Figure R2) and the river-ocean interaction (Figure R3) in the revised manuscript. Improvement on the numerical accuracy of storm surge level simulation is also emphasized. We added some detailed explanations in the revised manuscript based on the reviewer's comments [Page 13, Lines 266-269, Lines 275-278; Page 14, Figure 6; Figure S7 in supporting material].

**Comment 7.** Figure 6: The upper subplot should also have the label. The comparison of flood inundation here seems overly qualitative. Could the authors provide some quantitative results to clarify the issue? For instance, what is the observed inundation area? What is the simulated inundation area? Are there significant differences between the two?

*Response:*

Thanks for the comment. We added a label on the upper subplot. In the revised manuscript, we introduce two metrics to assess the performance of the numerical model on describing inundation [Page 15, Lines 296-300]. As shown in Figure R4 and Figures S8-S11, the numerical model leads to satisfactory results on inundation areas. Most of the flooded regions identified by remote sensing are identical with the simulated results. Since the cloud cover removing technique in Tellman et al.'s (2021) method is known to result in an underestimation of the flooding area, it is reasonable to expect that the computed inundation range exceeds the observed area to some degree. We add the discussion of the inundation simulation results in the manuscript. [Page 15, Lines 296-307; Page 16, Figure 7; Figures S8-S11 in supporting material]

[Figure]

Figure R4. The contour plot (blue) of simulated inundation depth in Typhoon Mangkhut. The orange dots show the inundation range estimated by MODIS datasets.

**Comment 8.** Figure 8: The quantitative results here should be reflected in the abstract.

*Response:*

Thanks for the comment. We have revised the abstract [Page 1, Lines 18-21].

**Comment 9.** Line 355: The conclusion is overly simplistic, with most of the section describing the simulation results in the Pearl River Estuary. It does not highlight the key points of this study, such as the effects of more comprehensive coupling and the improvements in simulation performance.

*Response:*

Thanks for the comment. We have revised the conclusion [Page 21, Lines 387-404].

**Reference**

Egbert, G. D. and Erofeeva, S. Y.: Efficient inverse modeling of barotropic ocean tides, J. Atmos. Ocean. Technol., 19, 183-204, https://doi.org/10.1175/1520-0426(2002) , 2002.

Garratt, J.: Review of drag coefficients over oceans and continents, Mon. Weather Rev., 105, 915-929, https://doi.org/10.1175/1520-0493(1977)105, 1977.

Large, W. and Pond, S.: Open ocean momentum flux measurements in moderate to strong winds, J. Phys. Oceanogr., 11, 324-336, https://doi.org/10.1175/1520-0485(1981)011, 1981.

Tellman, B., Sullivan, J. A., Kuhn, C., Kettner, A. J., Doyle, C. S., Brakenridge, G. R., Erickson, T. A., and Slayback, D. A.: Satellite imaging reveals increased proportion of population exposed to floods, Nature, 596, 80-86, https://doi.org/10.1038/s41586-021-03695-w, 2021.

Zhang, A. and Yu, X.: A Major Improvement of Atmospheric Wave Boundary Layer Model for Storm Surge Modeling by Including Effect of Wave Breaking on Air-Sea Momentum Exchange, J. Phys. Oceanogr., https://doi.org/10.1175/JPO-D-23-0233.1, 2024.

**Authors' Responses to Comments of Reviewer #2**

*We appreciate very much the constructive comments and suggestions of the reviewer and have revised the manuscript accordingly. In the following, we explain our response to each comment of the reviewer. All revisions are highlighted in red color in the marked manuscript.*

**General Comments:**

The manuscript develops a land-river-ocean coupled model to simulate compound floods caused by the combined effects of storm surges, astronomical tides, storm waves, precipitation, and river flow. The model's effectiveness is validated through hindcasting of five typhoon events (including Hagupit 2008, Koppu 2009, Vicente 2012, Hato 2017, and Mangkhut 2018) in the Pearl River Delta region. The manuscript explores the contributions of ocean processes, river base flow, and precipitation to compound floods, and compares the results of one-way and two-way coupling models, revealing the impact of different coupling approaches on flood simulation outcomes.

The study is well-structured, and the methodology is sound. Nevertheless, it is expected that the manuscript will undergo further refinement before publication to address the identified issues.

**Response:**

The authors are grateful to the reviewer for his/her conclusion that the study is well-structured, and the methodology is sound. We try our best to address the comments of the reviewer and improve the manuscript accordingly.

**Comment 1.** Although the manuscript highlights the advantages of the two-way coupling model, it does not address the computational cost comparison with one-way coupling. A discussion on the computational complexity, including whether two-way coupling significantly increases computation time and if optimization can reduce costs, would be

valuable.

*Response:*

Thanks for the comment. Introducing a physically more reasonable description of the air-sea interaction as well as the river-ocean interaction, the accuracy of the numerical results is very much improved. In addition, the accuracy of the numerical results becomes less independent on the location of the river-ocean boundary, or the robustness of the model is enhanced. The price of doing so is a more than doubled computational cost. As a future work, we shall try to further optimize the model to reduce computational time without a significant loss of accuracy and robustness of the model. In this study, we focus on the effectiveness of the coupled model while retaining an acceptable computational efficiency. We added some discussions in the revised manuscript [Page 21, Lines 387-404].

**Comment 2.** The manuscript mainly focuses on the Pearl River Delta region. Although it mentions the model's general applicability, it would be helpful to include a discussion in the conclusion on the model's applicability to other regions, such as large river deltas or low-lying areas, and assess whether the model needs to be adjusted based on the characteristics of different regions.

*Response:*

Thanks for the comment. The theories on which the model is based is generally valid. Major components of the model system have been widely applied to various problems and well verified. All of the data required in model, except for high-resolution bathymetry data of the delta region, are publicly accessible and globally covered. Thus, we believe that the established model potentially has a global applicability. When applied to other large river delta regions, the mesh size may need to be carefully adjusted considering a balance between numerical accuracy and computational coast. We added some discussions in the revised manuscript [Page 11, Lines 225-227].

**Comment 3.** The manuscript demonstrates the advantages of two-way coupling by comparing

it with one-way coupling. It would be beneficial to further compare this model with existing coupled models, particularly the most advanced ones, and discuss its strengths and limitations, especially in terms of accuracy and efficiency.

***Response:***

Thanks for the comment. An important improvement of the model is the description of the air-sea momentum exchange. In most previous studies, the wind stress in the storm surge model is estimated with linear formulas, which oversimplify the air-sea interactions (Gori et al., 2020; Du et al., 2024). In order to compare our model with existing models, we added results from numerical experiments that use the default wind stress formula in ADCIRC (Garratt, 1977). The maximum storm surge level is compared in Figure R1. It is evidently shown that the maximum storm surge levels obtained with the e-AWBLM are more accurate than the default method. The limitations of the proposed model are mentioned in our response to Comment 1. We added some discussions in the revised manuscripts based on the reviewer's comments [Page 13, Lines 266-269; Figure 6].

[Figure]

Figure R1. The simulated maximum surge levels are compared with observations in two methods: (a) Garratt (1977) and (b) e-AWBLM

**Comment 4.** LN 152-155: " In this study, we perform … simulate compound floods." - The original sentence is too long and contains too much information. It should be split into two

clearer and simpler parts to improve readability. - "This study hindcasts five tropical cyclones (TC) events [Hagupit (2008); Koppu (2009); Vicente (2012); Hato (2017); Mangkhut (2018)] that caused destructive floods in the Pearl River Delta over the past two decades, to validate the model developed for simulating compound floods."

*Response:*

Thanks for the kind comment. We have revised the relevant sentence [Page 7, Lines 156-159].

**Comment 5.** LN 358-359:" …induced by five Typhoon events occurred…" - "Occurred" is in the past tense, but since "induced" is in the past participle form, they should be consistent. Therefore, "occurred" needs to be changed to the past participle "that occurred" to maintain tense consistency in the sentence.

*Response:*

Thanks for the kind comment. We have revised the relevant sentence [Page 21, Lines 393].

**Comment 6.** The image quality is insufficient. It is recommended to replace the image with a vector format.

*Response:*

Thanks for the comment. We revised all figures and guaranteed their quality.

**Reference**

Du, H., Fei, K., Wu, J., and Gao, L.: An integrative modelling framework for predicting the compound flood hazards induced by tropical cyclones in an estuarine area, Environ. Modell. Softw., 105996, https://doi.org/10.1016/j.envsoft.2024.105996., 2024.

Garratt, J.: Review of drag coefficients over oceans and continents, Mon. Weather Rev., 105, 915-929, https://doi.org/10.1175/1520-0493(1977)105<0915:RODCOO>2.0.CO;2, 1977.

Gori, A., Lin, N., and Xi, D.: Tropical cyclone compound flood hazard assessment: From

investigating drivers to quantifying extreme water levels, Earth's Future, 8, e2020EF001660, https://doi.org/10.1029/2020EF001660, 2020.

Xu, Y. and Yu, X.: Enhanced atmospheric wave boundary layer model for evaluation of wind stress over waters of finite depth, Prog. Oceanogr., 198, 102664, https://doi.org/10.1016/j.pocean.2021.102664, 2021.